# Development and Applications of Compton Camera—A Review

**DOI:** 10.3390/s22197374

**Published:** 2022-09-28

**Authors:** Raj Kumar Parajuli, Makoto Sakai, Ramila Parajuli, Mutsumi Tashiro

**Affiliations:** 1Department of Molecular Imaging and Theranostics, National Institutes for Quantum Science and Technology, 4-9-1 Anagawa, Inage, Chiba 263-8555, Japan; 2Gunma University Heavy Ion Medical Center, Gunma University, 3-39-22 Showa-machi, Maebashi 371-8511, Japan; 3Tizig Pharma Pvt., Ltd., Naxal, Kathmandu 45210, Nepal

**Keywords:** Compton camera, detectors, γ-rays, medical imaging

## Abstract

The history of Compton cameras began with the detection of radiation sources originally for applications in astronomy. A Compton camera is a promising γ-ray detector that operates in the wide energy range of a few tens of keV to MeV. The γ-ray detection method of a Compton camera is based on Compton scattering kinematics, which is used to determine the direction and energy of the γ-rays without using a mechanical collimator. Although the Compton camera was originally designed for astrophysical applications, it was later applied in medical imaging as well. Moreover, its application in environmental radiation measurements is also under study. Although a few review papers regarding Compton cameras have been published, they either focus very specifically on the detectors used in such cameras or the particular applications of Compton cameras. Thus, the aim of this paper is to review the features and types of Compton cameras and introduce their applications, associated imaging algorithms, improvement scopes, and their future aspects.

## 1. Introduction

The concept of a Compton camera was initially proposed by Schonfelder in 1973 [1]. Subsequently, Compton cameras were tested for diverse applications, especially in high-energy astrophysics (as components of telescopes and satellites) [2,3,4,5,6,7], as well as for environmental radiation measurements, such as the detection of radioactive elements in soil and open air [8,9,10,11]. A Compton camera is a γ-ray detector that uses the kinematics of Compton scattering to reconstruct a γ radiation trajectory. γ-ray interactions with a material can be explained by three main mechanisms: photoelectric absorption, Compton scattering, and pair production. In photoelectric absorption, a photon is absorbed by an atomic electron, leading to the ejection of electrons from the orbitals of the atom. Compton scattering is the elastic scattering of incident photons by the electrons of a scattering medium and is dominant, especially, in the energy range of several tens of keV to a few MeV. In pair production, a photon is transformed into an electron–positron pair. The pair production phenomenon is dominant at energies > 10 MeV. The Compton effect, which is dominant in a wide energy range, as shown in Figure 1 [12], is central to one of the most promising γ-ray detection devices—Compton cameras—that achieve a high sensitivity by suppressing the background, as demonstrated by the imaging Compton telescope (COMPTEL) [4,13,14,15,16] onboard the Compton γ-Ray Observatory (CGRO) satellite. In the Compton scattering mechanism, the incident γ-ray transfers a fraction of its energy to an electron and is scattered at a certain angle with respect to its initial direction. The energy transferred by the photon during the scattering is converted to the kinetic energy of the secondary electron. Thus, an incident γ-ray source can be identified based on the Compton scattering information recorded in the detectors of a Compton camera. A fundamental Compton camera consists of two types of sub-detectors: scatterers and absorbers. Figure 2 shows a schematic of the fundamental Compton camera. An incident γ-ray source is identified by evaluating the successive interactions of the incident photons with the scatterer and absorber. The γ-rays emitted by an individual source undergo Compton scattering in the scatterer and are subsequently absorbed by the absorber. These scattering and absorption events are together called a *Compton event*. The cones, formed by the Compton scattering angle and the interaction position in the scatterer, are accumulated together to locate the γ-ray source. If we assume that the incident γ-ray deposits an energy *E*_1_ in the Compton scatterer and the remaining energy *E*_2_ in the absorber, then the total incident energy, represented by *E*_in_, is expressed by the following equations:*E*_in_ = *E*_1_ + *E*_2_(1)

The scattering angle *θ* is calculated as follows:cos *θ* = 1 − ((*m*_e_ *c*^2^ *E*_1_)/(*E*_2_(*E*_1_ + *E*_2_))(2)
where *m*_e_ *c*^2^ is the rest-mass energy of the scattered electron. Therefore, estimating the fine position and energy resolution are the key requirements for a Compton camera to retain a low background with Compton kinematics.

In the field of high-energy astrophysics, the investigation of polarized γ-rays, which are generated by high-energy phenomena such as synchrotron radiation generation, bremsstrahlung, and Compton scattering [17], has remained a challenging task owing to the unavailability of suitable detectors that can efficiently detect and measure these polarized γ-rays. Since the development of Compton cameras, they have emerged as viable devices that can detect the polarized γ-rays originating from high-energy astrophysical phenomena. The Compton cameras also provide information on the emission geometries and magnetic field configurations. After the introduction of Compton cameras in astrophysics, many telescopes have been proposed and tested for observing various celestial phenomena. Another accomplishment of the Compton camera in astronomy is the detection of the Crab Nebula in 2011 by a Ge detector-based Compton camera onboard a balloon-borne system [18]. Further, following the Fukushima Daiichi Nuclear Plant (FDNP) disaster in 2011, γ-emitting radioisotopes (RIs), mainly ^137^Cs, ^134^Cs, and ^131^I, were released into the atmosphere and remained in the environment. Some of these released RIs, and their emissions and half-lives are listed in Table 1 [19]. Accordingly, to date, various types of light and portable Compton cameras with increased sensitivities have been developed to identify such radiation hotspots for decontamination [9,20,21].

The booming success of Compton cameras in astrophysical observations has propelled their subsequent application in diverse fields, especially in medical imaging. Todd et al. [5] proposed Compton cameras as suitable alternatives to the previously used mechanically collimated imaging systems, such as Anger cameras [22], and inspired by this, Singh and Doria [23,24] developed the first working prototype of a Compton camera for medical imaging in the early 1980s. During the 1980s and 1990s, semiconductor-based Compton cameras were introduced, based on the scheme proposed by Singh et al. Later, in 1993, Martin et al. [25] proposed a ring Compton scatter camera composed of a high-purity Ge (HPGe) detector and a ring array of cylindrical NaI scintillators. Later, LeBlanc et al. [26,27,28] replaced the Ge detector with an arrayed Si pad detector to develop a Si pad detector-based Compton camera, namely C-SPRINT, for nuclear medicine applications. Compared with the mechanically collimated single-photon emission computerized tomography (SPECT) system, the γ-ray detection efficiency of the C-SPRINT was found to be higher at lower energies. In 1988, Kamae et al. [29] proposed the concept of a multiple scattering Compton camera composed of layers of thin Si strip detectors surrounded by a cylindrical CsI scintillator. Based on the design proposed by Kamae et al., Dogan et al. [30] proposed image reconstruction by multiple scattering of γ-rays. In 2004, Wulf et al. [31] developed a Compton camera using three layers of double-sided Si strip detectors (DSSDs) and reconstructed images and spectra produced by ^137^Cs and ^57^Co γ-ray sources using this multilayered Compton camera, based on the report of Kroeger et al. [32]. In 2007, Vetter et al. [33] developed a Si/Ge Compton camera by combining the superior features of the HPGe detectors and DSSDs. 

These reported studies indicate that Compton cameras have been actively used in medicine, especially in nuclear medicine and hadron therapy, in the past two decades. Nuclear medicine, which utilizes γ-ray emitting RIs to capture the physiological information of tumors, is a vital diagnostic tool. Nuclear diagnosis is performed by administering the RIs into the patient through the intravenous route, and the built-in surrounding detectors detect the γ-rays emitted from the tumors to reconstruct an image. Thus, to obtain precise information on the distribution of these RI tracers, high-precision γ-ray detectors are necessary. SPECT is one of the commonly used medical imaging techniques, in which single-photon (γ-rays) emitters are detected, and subsequently an image is developed based on the detected photons. The SPECT systems employed a mechanical collimator, which determines the spatial resolution and sensitivity of this imaging system and whose dimensions need to be changed according to the target organ, measurement time, and data collection method [34,35,36,37,38,39]. Additionally, only low-energy (~300 keV) tracers can be used in the mechanically collimated SPECT systems. Therefore, there is a high demand for alternative SPECT imaging modalities. As an alternative, Carminati et al. proposed and developed a silicon photomultiplier (SiPM)-based magnetic resonance imaging (MRI)-compatible SPECT system that passed through preliminary clinical imaging tests [40]. The detectors used in this system are thallium-doped cesium iodide (CsI(TI) coupled with SiPMs. The preliminary evaluation shows that the extrinsic spatial resolution was better than 10 mm and thus established itself as a promising alternative for the conventional SPECT. On the other hand, in the last decade, Compton cameras, which do not require mechanical collimators and can detect a wide range of γ-ray energies, have come under the spotlight as feasible alternatives to the mechanically collimated SPECT in nuclear medical imaging. Compton cameras facilitate simultaneous imaging of multiple RI tracers, yielding detailed information on lesions as well as enabling instantaneous identification of the elements corresponding to specific functions [38].

Recently, an interesting application of Compton cameras, i.e., online beam-range monitoring in hadron therapy, has materialized, because the energy of the secondary prompt γ-rays originating from the interaction of the beam with the cells in a patient’s body ranges from a few hundred keV to a few MeV. The utilization of prompt γ-rays for range verification was first proposed by Jongen and Stichelbaut in 2003 [41] and first realized by Min et al. in 2006 [42]. The range information can be extracted from the spectral, temporal, or spatial patterns of the prompt γ-rays [43,44,45]. A correlation exists between the emission point of the γ-rays and the Bragg peak position [46,47]. In hadron therapy, the verification of the stopping range of the irradiated protons or heavy ions in a patient’s body is critical to minimize the inherent range uncertainty, which is realized by detecting the emitted prompt γ-rays [48,49]. Although the efficiency of a Compton camera is superior to that of the Anger cameras [50,51], challenges such as low count rates, incomplete sampling, and unknown initial photon energies impede the effective implementation of these novel devices in medical therapy. Therefore, various studies are being conducted worldwide to improve the Compton camera efficiency, imaging algorithms, measurement techniques, and detector specifications for real-time dose verification in hadron therapy.

Very few review papers on Compton cameras exist in the literature; those that have been published only cover specific topics (e.g., the detectors used in Compton cameras or their specific uses [52,53,54,55]). Moreover, there is a lack of current review papers on Compton cameras and their applications. We believe that this paper will act as a source of knowledge on Compton cameras, their development history, their major fields of application, and the trends in imaging algorithms used in such cameras. The principle aim of this review is to introduce and explain the different types of Compton cameras according to the types of detectors used, measurement targets, and application fields. Further, the details of several popular image reconstruction methods for Compton cameras, reported by different authors, have been presented here. We also discuss the pitfalls of the Compton camera as well as the avenues for future improvements.

## 2. Types of Compton Cameras

### 2.1. Si/CdTe Detector-Based Compton Cameras

Generally, semiconductor-based scatterer detectors are widely developed because of their high detection efficiencies and excellent energy resolutions. Table 2 lists the most commonly used semiconductors in detectors and their properties [56]. Among the different types of detectors used in Compton cameras, Si/cadmium telluride (CdTe) detectors are considered suitable for γ-ray measurements. The Si/CdTe Compton cameras exhibit a high energy resolution as well as a high angular resolution, especially for γ-rays of a few hundreds of keV. Si detectors are characterized by excellent energy and spatial resolutions because of the lower photoabsorption cross-sections of Si as well as the smaller Doppler broadening, compared with other semiconductors [57]. In contrast, CdTe detectors, which have a high density and a high effective atomic number, comprise high-resolution Schottky diodes, thus their energy resolution and efficiency are better than those of the Si detectors [58,59,60,61]. The beginning of the twenty-first century was marked by the development of Si/CdTe Compton cameras, as successors to COMPTEL, by Takahashi et al. [62,63,64]. These cameras use multiple layers of DSSDs, CdTe pixel detectors, and a highly advanced low-noise analog application-specific integrated circuit (ASIC) [63]. Figure 3 shows actual photographs of the (a) stack of DSSDs, (b) stack of CdTe detectors, and (c) the prototype Si/CdTe Compton camera. Figure 3a,b were reprinted from the works of Watanabe et al. [65] and Figure 3c was reprinted from Takeda et al. [66]. In DSSDs, highly doped p-type and n-type Si strips are implanted orthogonally to provide two-dimensional coordinate measurements. Meanwhile, the CdTe pixel detector is composed of indium as a common electrode side and the other platinum electrode side is pixelated, in which a thin layer of gold is evaporated for etching. Initially, a Si/CdTe Compton camera was developed as a soft γ-ray detector and launched into the low Earth orbit onboard the Hitomi satellite for the ASTRO-H mission, previously known as the NeXT mission, [67,68,69,70] in 2013 as a successor to the Suzaku X-ray mission [71,72,73,74]. The Si/CdTe Compton telescope can be operated at a moderate temperature of 0 to −20 °C. Following the Fukushima Daiichi Nuclear Power Plant disaster in 2011, because of the Great East Japan Earthquake [75], a Si/CdTe Compton camera, ASTROCAM, was built [76] to measure the radiation contamination (hot spots) at the accident site and in the surrounding environment, including the soil around plants and trees [77,78]. The camera consists of eight layers of Si detectors and four layers of CdTe detectors of (size: 5 cm × 5 cm and weights: 8–13 kg). These Si/CdTe cameras exhibit good energy and angular (2.2% at half maximum (FWHM) and 5°, respectively) resolutions for photons with an energy of 662 keV, and a high-count rate of 0.16 cps/MBq at a distance of 1 m. The medical applications of the Si/CdTe Compton camera were introduced by Sakai et al. [79], especially in nuclear imaging of different RIs, followed by improvements in the Compton imaging algorithms [80,81,82]. The in vivo [83] and human imaging [84] applications of Si/CdTe Compton cameras are vital achievements of these devices in the field of nuclear medical imaging.

A Si scatterer has a low sensitivity to high-energy (several MeV) γ-rays. Therefore, the feasibility of using these detectors to detect the γ-ray emissions of energies lower than those used in other beam-monitoring applications is currently under study. Parajuli et al. [85] used ASTROCAM for the monitoring of the range of C-ion beams used in radiotherapy by measuring the 511-keV annihilation γ-rays and 718-keV prompt γ-rays [86]. Shiba et al. [87] performed an in vivo monitoring of the annihilation γ-rays released from a mouse irradiated by a C-ion beam. Turecek et al. [88] reportedly developed a Si/CdTe Compton camera based on the Timepix3 technology. The study demonstrated the benefits of using Timepix3 technology that could enhance resolution and reduces Compton camera size. The Compton camera was revised with the application of only a single layer detector of CdTe [89] and the combination of the Timepix3 detector with the miniaturized MiniPIX TPX3 readout system. The performance evaluation of the camera demonstrated its feasibility to distinguish gamma sources located at near and further distances from the detector and could differentiate the source type by their energy. Tomita et al. [90] proposed and developed a vehicle-mounted 4π Compton imager based on the 3D pixel array CdTe detector. The position and activity of a single ^137^Cs was estimated quantitively in 3D voxel space at three positions. Further, some other groups have assessed the feasibility of substituting CdTe with cadmium–zinc–telluride (CZT), which exhibits a comparatively higher resistivity and therefore lower leakage currents, to develop Compton cameras with suppressed background noise [91,92,93,94].

### 2.2. Ge Detector-Based Compton Cameras

Ge detectors are popular alternatives to Si detectors in Compton cameras because of the advantageous characteristics of Ge over those of Si; for instance, HPGe has an excellent energy resolution of approximately 0.2% at 662 keV [95]. However, Ge-based detectors must be operated at very low cryogenically-regulated temperatures, owing to their small bandgaps [96,97]. Moreover, efficiency calibration is essential to perform measurements using HPGe detectors [98]. Ge detectors, such as Gammasphere [99], Euroballs [100], and AGATA [101], use Compton-suppressed arrays, and as a result, exhibit excellent energy resolutions, high efficiencies, and high peak-to-total ratios, thereby effectively mitigating the shortcomings of the NaI detectors [102]. Singh et al. [23,24] showed that the single Compton scatter efficiency of Ge is higher than that of Si, and thus the front-detector thickness should be minimal (<10 mm) to achieve an acceptable spatial resolution. They replaced the mechanical collimator with a 6 mm × 6 mm Ge detector as the front detector and measured the γ-rays emitted by ^99m^Tc (140 keV) and ^137^Cs (662 keV) RI sources. Despite the resolution being poorer than expected, Ge-based front detectors are still preferred over Si detectors in Compton cameras for imaging photon energies in the wide range of 140–511 keV. An improved position-sensitive HPGe Compton camera, SmartPET (Smart Positron Emission Tomography), was developed and evaluated by Cooper et al. [103] and Boston et al. [104] at Liverpool University. The detector size was 60 × 60 × 20 mm^3^, the spatial resolutions were 7.7 and 6.3 mm for 511 and 1408 keV, and γ-ray emitting sources placed 50 mm away from the detector, respectively. To overcome this spatial resolution, Takeda et al. [105] developed a Ge-based hybrid Compton camera to image multiple probes. This detector showed a spatial resolution of 3.2 mm for an 834-keV γ-ray source placed 3.5 cm away from the camera. However, the efficiency decreased with the thickness of the mask, and the image reconstruction was more complex. Japan’s largest research institution, RIKEN, reportedly developed a Ge-based Compton camera, namely γ-ray emission imaging (GREI), for simultaneous and nondestructive imaging of radionuclides [106,107]. In this double-sided Ge detector, the active volumes of the scatterer and absorber are 39 × 39 × 10 and 39 × 39 × 20 mm^3^, respectively, and the strip pitch is 3 mm for both of the detectors. The detection efficiency of the GREI system for 662-keV γ-rays at a distance of 15 mm is approximately 0.01%. The GREI system was subsequently improved by narrowing the distance between the Ge detector elements (60–40 mm) and by increasing the data acquisition speed (GREI-II). The GREI and GREI-II systems were employed to evaluate and compare the distribution of bio-metals, such as ^64^Cu and ^65^Zn, using mouse models [38,108]. The research groups in RIKEN have renewed their efforts to implement GREI in clinical diagnostics for imaging multiple biological processes. Alnaaimi et al. [109] also developed a Ge-based Compton camera (Figure 4) for medical applications and achieved an angular resolution of 9.4° ± 0.4° for a 662-keV γ-ray emitting source.

### 2.3. Scintillator-Based Compton Cameras

Scintillator-based Compton cameras possess a high detection efficiency, are less expensive, and are operable at room temperature, although their angular resolution is relatively lower than that of the semiconductor-based Compton cameras because of poor energy and position resolution. Popular scintillators usually used in Compton cameras are thallium-doped sodium iodide (NaI(TI)), thallium-doped cesium iodide (CsI(TI)), sodium-doped cesium iodide (CsI(Na)), cerium-doped gadolinium aluminum gallium garnet (Ce:GAGG), and cerium-doped lanthanum bromide (Ce:LaBr_3_). The properties of scintillator detectors used in Compton cameras are listed in Table 3 [110,111]. Typically, scintillators are used with photomultiplier (PM) tubes or Si photodiodes. With the development of multipixel photon counters (MPPCs), which are a type of SiPMs, scintillators have become easier to use as scatterers in Compton cameras. Generally, the scintillation process occurs via three steps: the excitation of electrons, energy transfer of the excited or ionized electrons, and emission of fluorescence. Hofstadter et al. [112] discovered NaI(TI) in 1948 and it was the most widely used scintillator until the other scintillators were discovered. NaI(TI) and CsI(TI) possess a high sensitivity but an insufficient energy resolution in the low-energy range. Most radiopharmaceuticals used in nuclear imaging emit γ-rays with energies < 250 keV, such as ^99m^Tc (141 keV), ^123^I (159 keV), and ^111^In (171 and 245 keV), among which ^99m^Tc accounts for more than 50% of the RI-based medical diagnostics. Therefore, these scintillators are not suitable for Compton cameras used in medical applications, although they are still used in environmental radioactivity measurements [113,114,115]. In contrast, the properties of Ce:GAGG scintillators are superior to those of the other scintillators because they can emit Ce^3+^ photons with a 520 nm wavelength that originate from the 5d–4f transitions in Ce^3+^, possess high densities, and exhibit high light outputs, fast decay times, low self-radiation, and a good energy resolution [116,117]. Ce:GAGG exhibits a high stopping power and is neither hygroscopic nor self-emissive. The demands for Ce:GAGG-based Compton cameras rapidly increased after the Fukushima nuclear disaster in Japan in 2011 to measure the levels of different radioactive elements from a distance. Kataoka et al. [118,119,120] developed a portable and handheld Compton camera based on a Ce:GAGG scintillator and MPPC with a depth-of-interaction (DOI) capability. The prototype Compton camera is shown in Figure 5, which has been reprinted from Kataoka et al. [118]. The left photograph shows the compact form of the Compton camera and the right photograph shows its internal structure. This camera, which contained 10 mm thick 50 × 50 mm^2^ Ce:GAGG plates acting as both the scatterer and observer coupled with the MPPC, was initially used to detect radioactive elements [121]. The camera had an angular resolution of less than 10° for the source and 10 mm for the DOI configuration. Its angular resolution, measured at the FWHM, was approximately 8° at 662 keV. Since 2016, this group has been engaged in finding suitable methods to build a Compton camera for γ-ray imaging for particle therapy applications [122,123,124]. Kenichiro et al. [125] also developed a Compton camera, namely a Compton–PET hybrid camera, based on Ce:GAGG detectors and demonstrated simultaneous imaging with ^131^I and ^18^F RI sources. The Compton–PET hybrid camera consisted of two Ce:GAGG Compton cameras facing each other, with the target object at the central axis. Takahashi et al. [126] also reported on the development and performance of the stacked GAGG scintillator-based omnidirectional Compton imager for the objective of the rapid measurement of radioactive fallout, such as in nuclear accidents of the FDNP disaster. The three-dimensional position resolutions were estimated using the prototype Compton camera to evaluate the performance of several kinds of scintillators.

### 2.4. Electron-Tracking Compton Cameras 

A typical Compton camera can only determine the scattering angle and cannot fully reconstruct the trajectory of the detected γ-rays. In electron-tracking Compton cameras (ETCCs), the tracking of the recoil electron reduces the Compton circle to a point and a significant background reduction can be achieved, compared with the conventional Compton cameras [127]. Tanaka et al. [128] developed the first ETCC that could detect Compton recoil electrons using a gaseous micro-time projection chamber (µ-TPC) detector and a position-sensitive scintillation camera enclosing the µ-TPC. Figure 6 shows the conceptual structure (left) and an actual photograph (right) of the ETCC, reprinted from Kabuki et al. [129]. This ETCC allowed three-dimensional (3D) position identification of the detected electron. Compared with other detectors, gaseous detectors show a better recoil–electron-tracking performance because of reduced multiple scatterings, although their detection efficiency is poor [130]. Muichi et al. [130] used various types of detectors as absorbers, one of which had a front detector sized at 10 × 10 × 8 cm^3^, a TPC (ethane) as the scatterer, and a 6 × 6 × 13 cm^2^ gadolinium orthosilicate (GSO) scintillator array as the absorber [129]. The angular resolution of the ETCC was 6.6° at 360 keV for ^131^I. Subsequently, the angular resolution was improved to 4.2 ± 0.3° at 662 keV by replacing the GSO scintillator by a LaBr_3_ scintillator [131]. Even though the efficiency and spatial resolution of the ETCC were lower than those of the other conventional Compton cameras, the high signal-to-noise ratio compensated for the low efficiency and resolution. They also reported simultaneous imaging of ^131^I and fludeoxyglucose (FDG) injected into mice [132]. After 2010, Tanimori et al. [133] and Mizumoto et al. [134] developed improved ETCCs and evaluated their astronomical and environmental γ-ray detection performances by changing the TPC size and gas parameters. However, these reported studies were more focused on the astronomical applications than on the nuclear medical imaging applications of ETCCs. To improve the detection efficiency, research is also being conducted to visualize the electron trails within solid-state detectors. The performance of Compton cameras would be greatly improved if the electron tracks in semiconductor or scintillator detectors could be visualized [135,136,137]. Jiaxing et al. [138] also reported the electron tracking algorithm using the Timepix3 based Compton camera, as Timepix3 possess an advantage of measuring the time and energy of an event simultaneously in each pixel. The demonstrative experiment result showed the significant enhancement of the angular resolution and the feasibility of the electron track algorithm. Yoshihara et al. [139] reported the development of a Compton imaging system that could track recoil electrons by using a combination of a trigger mode silicon-on-insulator (SOI) pixel detector and a gadolinium aluminum gallium garnet (GAGG) detector. The experimental results showed that the coincidence events were detected at a maximum rate of 1 cps for the measurement of 662 keV γ-rays of ^137^Cs. They remarked that their Compton camera is suitable for imaging nuclides that emit 100–300 keV γ-rays.

### 2.5. Other Compton Cameras

In addition to the abovementioned Compton cameras, other Compton cameras based on semiconductor detectors and scintillators are under development. To counter the low sensitivities and high costs of Si-based Compton cameras, Katagiri et al. [140] developed a europium-doped calcium fluoride (CaF_2_(Eu)) scintillator-based omnidirectional Compton camera for environmental radiation monitoring in nuclear medicine facilities. This CaF_2_(Eu) scintillator-based Compton camera could detect γ-ray sources emitting energies < 250 keV. Four CaF_2_(Eu) crystals, each with a diameter of 2.54 cm, were set as the vertices of a tetrahedral-structured scatterer and absorber. These Compton cameras could reportedly suppress ghost images and they exhibited a 12° angular resolution for ^57^Co γ-rays, as well as better detection efficiencies.

Kasper et al. [141] developed a SiPM and a scintillation fiber-based Compton camera. Both the scatterer and absorber consisted of a thin elongated fiber made of a high-density inorganic scintillator (Ce:LYSO, Ce:LuAG, and Ce:GAGG) coupled with a SiPM. They tested the feasibility of using this Compton camera for beam monitoring in proton therapy and found that the camera with a Ce:LYSO scintillator exhibited a good proton beam monitoring performance.

Barrientos et al. [142] is currently working on the development and updating of LaBr_3_-coupled SiPM-based Compton cameras (MACACO II) for proton beam monitoring. The MACACO II exhibits an energy resolution of 5.6% (FWHM) at 511 keV and an angular resolution of 8°.

Another popular Compton camera is the Polaris J^TM^ developed by H3D. Polf et al. [143] evaluated the detection (imaging) performance of this Compton camera, which is based on CZT detectors because of their high γ-ray interaction cross-sections for 6 MeV γ-rays. Polaris J consists of four stages, and each stage consists of a separate Polaris detection system containing a 20 × 20 × 15 mm^3^ CZT detector. The CZT detectors are pixelated in an 11 × 11 pattern on the *x* and *y* anode sides. The energy resolution of the Polaris J system is 9.7 keV (FWHM) at 662 keV (emitted by ^137^Cs). This system is being further improved for proton therapy applications.

The Brookhaven National Laboratory, in collaboration with the National Aeronautics and Space Administration, is also developing a CZT-based Compton camera for astronomical observations [144]. The detecting plane is an array of 8 × 8 × 32 mm^3^ position-sensitive virtual Frisch-grid CZT detectors in the shape of bars and can detect the γ-ray interaction points better than any other CZT detector. Each module is a 4 × 4 detector subarray coupled with an ASIC. The detector exhibits a less than 1% energy resolution at 1 MeV and a submillimeter position resolution. The feasibility of this Compton camera is under study; it is expected to be sent to higher altitudes through balloon flights for sensitive measurements.

## 3. Applications of Compton Cameras

### 3.1. Astronomical Observations

In the recent years, most of the research on Compton cameras has been focused on medical applications. However, Compton cameras were originally introduced and developed for γ-ray detection in astronomy, and such Compton cameras were initially known as Compton telescopes. In γ-ray astronomy, the detection of both low- and high-energy γ-rays (up to 1 MeV and 1–10 MeV, respectively) is challenging owing to low photon signals, contaminated backgrounds, and the complex Compton scattering process that occurs in the detectors. In astrophysics, precisely locating the position of the γ-ray source as well as measuring the emitted γ-ray energy are pivotal for exploring unobserved celestial objects such as black holes, γ-ray bursts, supernova remnants, γ-ray pulsars, active galactic nuclei, and numerous other known and unknown objects [145,146]. The first Compton camera, COMPTEL, onboard the CGRO was a scintillator detector-based Compton camera, which facilitated the first complete all-sky survey in the energy range of 0.75 to 30 MeV [147], and the highest survey sensitivity was obtained in the energy range of 1–10 MeV. The number of most significant detections was 32 for steady sources and 31 for γ-ray burst sources. The launch of COMPTEL and its encouraging results propelled the development of new Compton cameras based on different detectors for measuring the γ-ray energies that are common to astronomical events.

The ASTRO-H was another Compton camera onboard the Hitomi satellite, which was launched from Japan on 17 February, 2016 [7] with the primary mission of detecting soft γ-rays using a Si/CdTe Compton camera. The Compton camera was composed of 32 layers of 0.625 mm thick Si pad detectors and eight layers of 0.75 mm thick CdTe pad detectors [7]. According to the preliminary information, it measured the soft γ-ray through reconstruction of Compton scattering covering an energy range of 60 to 600 keV with high sensitivity (10 times better than Suzaku) [7]. Unfortunately, the Hitomi satellite lost contact with the ground on 26 March 2016; thus, the sufficient planned measurements were unsuccessful.

As another approach to MeV γ-ray measurement, the sub-MeV γ-ray imaging loadedon-balance experiment (SMILE) project based on the use of an ETCC developed by Tanimori et al. was implemented to measure diffused γ-rays. SMILE-1, launched in 2006 with a detector size of 10 × 10 × 15 mm^3^, successfully detected diffused cosmic and atmospheric γ-rays and performed powerful background rejection [148,149]. The recent model, SMILE-2+, had a successful balloon flight in April 2018 in Australia [150]. The effective area of the ETCC was 1.1 cm^2^ for 0.356 MeV, and the point spread function (PSF) had a 30° half-power radius (HPR) of 0.662 MeV. It had a sensitive detector volume of 30 × 30 × 30 mm^3^ and was filled with argon mixed gas (Ar: CF_4_: iso C_4_H_10_; pressure ratio = 95:3:2) at a pressure of 2 atm. GSO pixel scintillator arrays, each containing 8 × 8 pixels with a size of 6 × 6 mm^2^, were adopted as γ-ray absorbers. SMILE-2+ ETCC recorded 4.9 × 10^7^ events after it was switched on. The detection sensitivity of ETCC was found to be the same as that at the ground level. Thus, there is a possibility that ETCC could become a sophisticated astronomical Compton camera for deeper all-sky surveys.

Another project is underway for a detector that can measure higher energies using the principle of the Compton camera and pair production. The project is named e-ASTROGAM (enhanced ASTROGAM), which is considered to be a breakthrough observational space mission led by joint research teams of more than 100 institutes worldwide [151]. The mission is expected to provide important information, such as LIGO-Virgo-GE0600-KAGRA, SKA, AMLA, E-ELT, LSST, JWST, ATHENA, CTA, and LISA. For instance, part of the 1–30 MeV region obtained in the 1990s (Figure 7; upper left), as illustrated by the simulated Cygnus region (Figure 7; lower right), is expected to be obtained from the e-ASTROGAM mission. The main constituents of e-ASTROGAM are trackers made of 56 planes of double-sided Si strip detectors of a 1 m^2^ area, a calorimeter made of an array of CsI(Tl) bars of 5 × 5 × 80 mm^3^ each having a relative energy resolution of 4.5% at 662 keV, and an anti-coincidence system composed of a plastic scintillator. The space information was gathered and analyzed.

### 3.2. Nuclear Medical Imaging

The use of a Compton camera for nuclear medical imaging has been of great interest in recent decades because of its higher sensitivity than collimated cameras and its ability to simultaneously image γ-rays of different energies over a wide energy range. A wide field of view can be achieved using a small Compton camera detector. PET and SPECT are established molecular imaging systems in nuclear medicine and have made a significant contribution to the evaluation of the physiological function and biochemical changes of molecular targets. PET imaging is based on the detection of radiation emitted from a positron-emitting radioactive tracer that is injected into the body. The annihilation γ-rays from radioactive tracers reach the detector (usually scintillators coupled with photodetectors) of PET, and thus imaging is performed. Although several isotopes emit a positron, simultaneous imaging of multiple tracers is difficult because the energy is single (511 keV). On the other hand, SPECT imaging is based on the detection of γ-radiation-emitting RIs, such as the isotopes of Tc and I. The γ-rays emitted from the body at different angles were captured by the SPECT detector, which was attached to the collimator. Because the directivity is obtained using a collimator, the energy range where imaging can be performed is limited by the thickness of the septum (collimator wall). To achieve a high resolution, the collimator interval should be reduced; however, because the incident photon is limited, there is a trade-off between the sensitivity and spatial resolution. To measure high-energy γ-rays, the septum must be thicker, further reducing the detection efficiency. Although the spatial resolution of Compton images has not yet been achieved like that of PET and SPECT, the theoretical facts, simulation studies, and experimental results show that the Compton camera and its imaging technique could be the best alternative to address the shortcomings of PET and SPECT. In nuclear imaging, depending on the type of diagnosis, it is necessary to image the isotopes of radioactive tracers, such as ^99m^Tc, ^123^I, ^18^F, ^111^In, ^201^Tl and so on, which possess different half-lives and energies. Some of the popular RI tracers used in nuclear medicine are listed in Table 4, along with their half-lives, decay type, and the energy of their γ-rays [52]. A Compton camera should possess the ability to image tracers with a high efficiency and sensitivity. The first Compton camera designed especially for nuclear medicine was a Ge detector-based Compton camera by Singh and Doria [23,24], which successfully demonstrated the imaging ability of ^137^Cs and ^99m^Tc [152], although the shape of the sample used in the experiment was simple and significantly different from that used in clinical settings. Since then, many studies have been conducted using different types of Compton cameras for application in nuclear medicine.

Several in vivo imaging studies have been conducted [107,153,154,155,156]. One of the progressive studies for in vivo imaging displayed the use of a Si/CdTe Compton camera, which was the first trial of a Compton camera in vivo [154]. Iodinated (^131^I) methylnorcholestenol and strontium chloride (^85^SrCl_2_) were injected into mice, and ^131^I and ^85^Sr, which release 364 keV and 514 keV γ-rays, respectively, were successfully measured by the prototype camera. Although a spatial resolution of a few millimeters was obtained, this study inspired the improvement of the detection efficiency of the Compton camera and the development of 3D imaging. As a continuation, the in vivo 3D imaging of ^131^I, ^18^F-FDG (511keV), and ^67^Ga-citrate (300 keV) was conducted [157]. Sakai et al. [79,80,81,82] at the Gunma University Heavy Ion Medical Center (GHMC) tested a Si/CdTe Compton camera, and successful measurements with ^99m^Tc and ^18^F within human phantoms were demonstrated in their studies. Finally, the in vivo results [83] in mice followed by human experiments [84] using a Si/CdTe Compton camera have been studied by researchers for its application in the medical field. Thus, the Si/CdTe Compton camera shows potential for the simultaneous imaging of two radiopharmaceuticals (^99m^Tc and ^18^F) in humans. Figure 8a,b shows the human experiment setup and the spectrum obtained from the Compton camera, reprinted from Nakano et al. [84]. Figure 8c,d show the Compton images overlaid with the CT images for DMSA and FDG imaging. Although the spatial and contrast resolution of the Compton camera still needs to be improved to surpass the abilities of PET and SPECT, the latest Compton camera has improved far more than that which was initially made.

Owing to the individual advantageous features of the Compton camera and PET imaging system, a synchronized Compton camera with a PET detector system is another application of the Compton camera in nuclear medicine which is of great interest. Shimazoe et al. [158] of the University of Tokyo, in collaboration with Tohoku University and the National Institutes for Quantum Science and Technology (QST), have recently developed a Compton–PET hybrid imaging system. The Ce:GAGG scintillation-based PET system and Ce:GAGG scintillator-based Compton imaging system were combined to develop a Compton–PET imaging system for simultaneous imaging. PET nuclide visualization using coincidence detection of annihilation γ-rays and the SPECT nuclide visualization technique were implemented. The feasibility and performance of the Compton–PET imaging system were also demonstrated using in vivo imaging in mice [156]. Figure 9 shows the reconstructed Compton images superimposed with computed tomography (CT) images to visualize ^111^I, and ^18^F-FDG accumulation in mouse body parts such as the liver, heart, and bladder using the Compton–PET imaging system.

Although the RI tracers in a mouse was imaged successfully, the hybrid Compton–PET still lags sufficient spatial resolution compared with small-animal PET scanners. The authors believe that the system can be upgraded to improve the resolution in terms of the detector pixel size, method of image reconstruction, and the design of imaging modalities such as ring-type Compton–PET. They are also working on whole γ-ray imaging, which is a combination of PET and Compton imaging, based on a dual detector-ring structure [159]. Nonpure positron-emitting nuclides, such as ^44^Sc (half-life = 3.97 h) and ^124^I (4.176 d), emit not only positrons but also additional γ-rays almost simultaneously. For each decay, the source position can be directly identified as the intersection of the surface of the cone given by Compton kinematics and the line given by the coincidence measurement of two annihilation γ-rays. Several research groups have reported triple γ-ray imaging or β+-γ coincidence simulation studies for these nonpure positron emitters [160,161,162].

Kataoka et al. [163] of Waseda University developed a compact Ce:GAGG scintillator and an MPPC-based Compton camera, particularly for molecular imaging. The developed Compton camera had typical energy and angular resolutions (FWHM) of 7.4% and 4.5°, respectively, for 662 keV. The prototype was successful in imaging ^137^Cs (662 keV) with the point source kept at 4 cm with a spatial resolution of 3.1 mm FWHM. In addition, ^22^Na (511 keV), ^137^Cs, and ^54^Mn (834 keV) point sources were simultaneously imaged using a 3D multicolor modality. Three-dimensional in vivo multicolor imaging in a mouse was also performed to image ^131^I, ^85^Sr, and ^65^Zn (1116 keV), which demonstrated that the tracers were correctly accumulated in each target organ [155]. ^131^I, ^85^Sr, and ^65^Zn RI tracers were injected into mice and 3D images were obtained using the Compton camera, as shown in Figure 10c–f. Figure 10a illustrates the detector alignment for 3D imaging, and Figure 10b Shows the energy spectra of ^131^I, ^85^Sr, and ^65^Zn RI tracers injected in mice. Although the spatial resolution due to the energy uncertainty was reported to be worse than that of SPECT and PET in small animals, the performance of the camera was believed to be comparable to that of SPECT. The research team is improving the Compton camera by developing a hybrid Compton camera that can simultaneously achieve X-ray and γ-ray imaging by combining the features of the Compton camera and a pinhole camera in a single-detector system [164].

Owing to the Compton camera principle, the spatial resolution of low-energy γ-rays is poor. Therefore, the use of high-energy γ-ray sources, which are not used in conventional SPECT with an Anger camera, has been proposed [51]. Unlike in astronomy, in nuclear medicine the measurement target is nearby and its radioactivity is very high. It is also characterized by the fact that the energy of the source is known. Therefore, it is necessary to develop a detector system that matches these characteristics.

### 3.3. Beam-Range Monitoring in Particle Radiotherapy

In comparison with its application in nuclear medical imaging, there is much ongoing research on the feasibility of the application of Compton cameras in particle radiotherapy, especially in proton therapy [47,141,165,166,167,168]. None of the research may be addressed by this review paper; however, some recent studies will be reported in this paper. Generally, particle radiotherapy is categorized into two categories: proton therapy and heavier positive ion therapies, especially carbon ion radiation therapy (CIRT). In all cases of particle therapy, it exploits the energy deposition phenomenon of ion beams, with the Bragg peak at the end of the range, to accurately irradiate tumors and minimize the unwanted dose to healthy tissues. However, owing to the lack of a well-established online beam-range monitoring/verification system between the planned and delivered treatment, the full exploitation of its clinical potential is limited. The uncertainties in the estimated range calculated from the CT images used for treatment planning were caused by anatomical changes during treatment. Whenever the beam interacts with the target, a wide energy range of γ-rays is emitted as secondary radiation that bears information on its location and is correlated with the Bragg peak position [42,49]. PET is one of the established methods; however, the images are affected by the metabolic-washout effect. Due to the long half-lives of the RIs, some parts of the positron emitters traveling far away from the origin point, and high-energy positrons via pair production are less related to the beam range [169,170]. For this reason, many research groups have focused on range verification using prompt γ-rays and hence using a Compton camera owing to its benefits, as explained in the introduction section.

As a first attempt, Polf et al. in 2015 [167] experimentally measured prompt γ-rays emitted by clinical proton beams irradiated on a water phantom. The Compton camera used in the study was Polaris J to measure the prompt γ-rays from 0.2 MeV to 6.5 MeV. Pencil beams of 114 MeV and 150 MeV and up to 500 cGy were used for irradiation. The study concluded that the emission profile that lined up with the distal Bragg peak fell with an accuracy of 1.5 mm for a 3 mm shift of the target. The study opened the door for practical tests of the Compton camera for beam-range monitoring, however, the dose used in the study was far higher than that used in clinical treatment. The group continues their study with the Polaris J Compton camera to develop 3D images and to conduct clinical trials [171]. A benchmark experiment was performed using a Tandetron accelerator at Helmho-Zentrum Dresden-Rossendorf (Germany). A two-stage Compton camera based on a CZT (scatterer) and BGO (absorber) was used to measure 4.44 MeV prompt γ-rays emitters due to the interaction of a proton beam of 0.9 MeV with stainless steel equipped with a Ta plate with TiN sputtered on the top [172]. This study proves the feasibility of imaging the localized 4.44 MeV prompt γ-rays rather than its clinical scenario. Heuso-Gonzalez et al. in 2017 [173] measured the feasibility of a block Compton camera (two stage BGO detector-based Compton camera) using a pencil-proton beam. The influences of the target shifts, target thickness, and beam energy varieties were evaluated. The study mainly recommends the use of high-density materials to maximize the coincident efficiency. Taya et al. [122] of Waseda University demonstrated the first real-time γ-ray-imaging ability of a handheld Compton camera for proton irradiation. A Ce:GAGG Compton camera was used to measure γ-ray emissions due to the irradiation of a 70 MeV proton beam on the water, calcium hydroxide (Ca(OH)_2_), and PMMA phantom. Both online and offline measurements were performed. Peaks of 511 keV γ-rays were observed in the energy spectra in both the online and offline cases. A 718 keV peak was observed during online irradiation. The study results could not confirm whether the 718 keV peak traces the Bragg peak. Subsequently, a 3D position-sensitive Ce:GAGG Compton camera was used to measure the 4.4 MeV prompt γ-ray energy range due to the 70 MeV proton beam on a PMMA phantom [123]. The improved Compton camera had an angular resolution of 5 degrees FWHM at 4.4 MeV and was able to discriminate multiple-Compton and escape events, which were the factors for background noises. The study reported that the 4.4 MeV prompt γ-rays has an intense peak near the Bragg peak (in agreement with the simulation results); however, the 511 keV prompt γ-ray image did not trace the proton dose (the result was against the suggestion of the simulation results).

Most studies have reported the use of proton beams, although CIRT is a more effective and sophisticated radiotherapy treatment modality. Parajuli et al. [85] used a clinical carbon ion beam to study the real-time measurement of the 511 keV annihilation γ-ray emissions from a PMMA target using the Si/CdTe Compton camera (ASTROCAM) [76], which is probably the first experimental study using a Compton camera in conjunction with CIRT. The Compton camera performance was evaluated by shifting the target with a water equivalent thickness of a few centimeters. The authors claimed that the Bragg peak was well-traced by the peak intensity position of the γ-ray emission position obtained through Compton images and was consistent with the simulation results. Shiba et al. in 2020 [87] performed an in vivo study using a mouse for the measurement of carbon ion-induced annihilation γ-rays using ASTROCAM. Figure 11 shows the experimental setup of the in vivo imaging of a mouse using ASTROCAM and the reconstructed Compton image of 511 keV annihilation gammas [87]. Positron emitter transport was observed by evaluating the range of γ-ray emissions after C-ion irradiation of the mouse abdomen. Using the same prototype camera, Parajuli et al. [86] again in 2021 reported the measurement of 718 keV prompt γ-rays released from a graphite phantom due to C-ion irradiation under clinical conditions. Theoretical and simulation analyses show that ^12^C collides with the target nuclei and produces ^10^B^*^ which is responsible for the generation of 718 keV prompt γ-rays. The study shows the preliminary feasibility of using the Si/CdTe Compton camera in CIRT, although the Compton camera requires adequate efficiency improvement. Figure 12 shows (a) the on-beam experiment setup for beam-range monitoring, (b) Compton image of 511 keV annihilation gammas, and (c) the Compton image of 718 keV prompt gammas. The figure has been rearranged and reprinted from Parajuli et al. [85,86].

All the experimental studies for particle therapy yet published are only phantom studies, except for the study conducted by Shiba et al. [87], as it is obvious that the application of the Compton camera for in vivo range monitoring is not yet ready. There are several candidates to be measured, but there is also a lot of background noise such as scattered rays and bremsstrahlung X-rays. Compton cameras, if they are to be implemented for particle therapy, should have the ability to tolerate the extreme load during dose delivery, the requirement of high-speed ASIC, a high performance even in low valid events, and an effective performance in a low dead time ratio.

### 3.4. Environmental Measurement

One of the necessary measures for nuclear power plants is to ensure radiation leakage in the environment by timely monitoring and to ensure radiation exposure to workers of the nuclear plants. According to the rules of the International Atomic Energy Agency [174], nuclear facilities worldwide should maintain some degree of decommissioning. Localizing the spatial location and type of radioactive source is a crucial step. More concerns and awareness regarding environmental radiation monitoring have increased since the nuclear accident that occurred at the Fukushima Daiichi Nuclear Plant (FDNP) in 2011. Various short-term and long-term effects of nuclear accidents have been reported by Matsunaga et al. [175]. Attempts are being made to implement different γ-ray detectors for environmental monitoring. For instance, as a recent progress, Carminati et al. [176] demonstrated the feasibility of a wireless, compact gamma-ray spectrometer aiming for the spotting of radioactive material within the metal scraps. The detectors used in the Compton camera were the NaI (Tl) scintillators combined with SiPM, that had an energy range from 60 keV to 1.5 MeV and was compatible with small magnetic fields. Three types of γ-ray detectors are available for localizing γ-ray-emitting radioactive sources: pinhole cameras, coded aperture γ-ray cameras, and Compton cameras. The detection efficiency of a pinhole camera is limited because of the geometrical area of the pinhole, whereas a coded aperture camera requires more detection efficiency for low-level polluted areas with a contamination below 1 µSv/h. Studies have shown that the Compton camera is the best alternative.

ASTROCAM 7000 HS was the first commercial soft γ-ray detector based on Si/CdTe detectors. Its feasibility for environmental monitoring was tested in an evacuation zone situated 20 km from the FDNP. The prototype camera was portable, containing two layers of Si detectors and three layers of CdTe detectors sized at 3.2 × 3.2 cm^2^ and a 250 µm pitch. ^133^Ba (356 keV), ^22^Na (511 keV), and ^137^Cs (662 keV) were individually measured, thus possessing a good detection efficiency of 1.68 × 10^−4^ (effective area 1.7 × 10^−3^ cm^2^) and an angular resolution of 3.8° obtained from stacks of five detector modules [66]. The efficiency achieved for ^137^Cs (662 keV) is 0.035 cps/MBq at 1 m. In later years, the prototype Compton camera was revised to increase the detector area to 5 × 5 cm^2^, the Si detector layer was increased to eight layers and CdTe was increased to four layers, and thus the efficiency of 0.06 cps/MBq at 1 m for ^137^Cs (662 keV) was achieved [80]. The first application of ETCC in environmental γ-ray imaging was reported by Tomono et al. in 2013 [177]. Contaminated soil (0.02 µSv/hr) bags of Fukushima were successfully measured, and fine images of radioactivity were obtained using ETCC. After revising the ETCC, a feasibility study of ETCC was again conducted using lab γ-ray point sources [134]. Clear images of ^137^Cs (662 keV) point source for the polar angles of 0–5° were obtained. Weak γ-ray source images of ^133^Ba (356 keV), ^137^Cs (662keV), and ^54^Mn (835 keV) were also successfully reconstructed, and the source positions are also clearly shown. The use of a Compton camera for environmental monitoring is costly because the Compton camera is complex and the employed detectors are expensive. On the other hand, considering the health and safety issues seriously, detection of the radioactive materials remotely is an essential measure. Very few studies could be acknowledged considering both the cost and remote sensing issues. Buonanno et al. [178] reported the development of a gamma-ray detection module that includes the facility of a drone-based localization of radioactive sources in the environment and uses a machine learning-based imaging technique for directional sensitivity measurement. While in the case of the Compton camera, to minimize the cost, many scintillator-based Compton cameras have also been used in practical applications. Jianyong et al. [179] reported the development and application of the GAGG (32 Ce:Gd_3_(Al,Ga)_5_O_12_) scintillator-based Compton camera. In this Compton camera, 4 × 4 array GAGG crystals were coupled with the 16 silicon photomultipliers and 16 avalanche photodiodes as the scatterer and absorber. Another striking feature of this development is that the Compton camera could be mounted on an unmanned helicopter for measuring ^137^Cs and ^134^Cs RIs which were released due to the accident in the FDNP, where people cannot enter due to the radiation causing health issues. Yoshiaki et al. [180] later improved the detection efficiency and angular resolution by increasing the 4 × 4 GAGG array to an 8 × 8 GAGG array. Measurements were performed over the FDNP surrounding areas. The results showed the accurate ambient dose equivalent rate maps at a height of 1 m with an angular resolution of approximately 10 m from a height of 10 m.

A Compton camera with a Ce:GAGG detector with the specifications explained by Kataoka et al. [118] was used to measure radiation contamination in the FDNP areas. The camera was redesigned to improve its portability and ability to perform remote imaging, and it can be mounted on a drone (Figure 13) [181]. Figure 13a shows the prototype drone carrying the Ce:GaGG Compton camera, while Figure 13b shows the stick PC used for communication with the optical camera. Figure 13c shows the optical image and Figure 13d shows the Compton image superimposed with the optical image of the region nearby the FDNP that contains three hotspots of radiation sources. The dose rates of the three hotspots were successfully measured. Compton cameras using CsI scintillators have also been studied and reported for environmental radiation measurements [9,113,115].

### 3.5. Other Specific Applications

In addition to the application of Compton cameras in astronomical measurements, medical imaging, and environmental measurements, there are ongoing studies on its application in indoor radiation measurements. Radiation measurements, such as inside the nuclear power plants, inside the RI facility, inside the PET imaging rooms, homeland security, and visualizations of naturally occurring radioactive materials (NORMs) in gas, oil, and metal factories, are critical for maintaining the health of employers working inside such facilities to monitor their exposure to radiation. Therefore, the application of the Compton camera is of interest. According to the latest report, Muraishi et al. [115] used an improved omnidirectional Compton camera to measure radiation in three different places: the RI facility, PET imaging room, and outside their private home near the FDNP. In this study, CsI (TI) and NaI (TI) scintillator-based Compton cameras were used. The prototype Compton camera successfully determined the position of the γ-ray source and the counting rate at a given position in the RI facility. In a PET facility, the possibility of visualizing the moment of radioactivity, such as the moment of ^18^F-FDG injected movement, has been reported. In addition, they claimed that their Compton camera could measure a surface dose of 1 µSv/h. In another of the latest studies, it was reported that the previous CsI scintillators were replaced with the CaF_2_(Eu) scintillator to improve the detection efficiency [140]. In another study conducted by Yuki et al. [182,183] of JAEA, radioactive contamination inside an FDNP facility was monitored. Decommissioning work is underway in the FDNP; therefore, it is very important to know the contamination inside and the dose rate distributions in the work environment to reduce worker exposure and ensure decontamination. The onsite measurements were successfully conducted by locating the hotspot in a sophisticated 3D image developed using a Compton camera, scanning laser range finder, and photogrammetry. The Compton camera used in this study was the Ce:GAGG Compton camera.

The application of a Compton camera for national or homeland security has not been reported in large numbers. There are many cases where the country is facing potential nuclear or radiological threats that could be controlled if a highly efficient and sensitive passive radiation source detector was developed [184,185]. Vetter et al. [33,186] used position-sensitive Si(Li) and HPGe detector-based Compton cameras for the passive detection of nuclear materials. Sweeney et al. [187] used a SmartPET Compton camera that can measure energies between 600 and 1500 keV γ-rays for homeland security. Alexis et al. [188] also reported the application of a Compton camera aimed at homeland security. This study investigated the performance of a large-scale two-plane scintillator detector-based Compton camera using different imaging algorithms on simulated data.

One of the planned applications of the Compton camera is the visualization of NORMs. Oil and gas field workers are particularly exposed to NORMs. It was announced that the Japan Oil, Gas and Metals National Corporation (JOGMEC), in collaboration with the Japan Aerospace Exploration Agency (JAXA) and the Mitsubishi Heavy Industries (MHI), is carrying out research and development of Compton cameras suitable for visualizing NORMs [189,190].

## 4. Image Reconstruction Methods for Compton Imaging

In γ-ray imaging, not only is the detector performance important, but the reconstruction algorithms also play a very important role, especially because Compton imaging is a complex computational task even for 2D imaging. The development of Compton cameras with complicated detector geometries leads to complex image-reconstruction processes. Different approaches, such as simple back-projection, analytic, and iterative methods, are being studied to achieve the best solution for image reconstruction in Compton imaging. Though there are many papers reporting on the image qualities, there are very few fair comparisons of the computational time among the imaging algorithms. This is because all imaging algorithms which are very complex should be developed for the same individual Compton camera and its external hardware used. Some image reconstruction approaches are explained below.

### 4.1. Simple Back-Projection Methods

The simple back-projection (SBP) method is a fast and straightforward algorithm used in the reconstruction of images in many imaging modalities. SBP is a necessary process in the filtered back-projection (FBP) and expectation-maximization (EM) algorithms, which are reviewed in the following sections. The SBP method is an important part of the FBP and EM algorithms and affects the image quality of each algorithm.

A Compton cone is reconstructed from the vector joining two interaction points, and the scattering angle is calculated from the Compton kinematics, as shown in Equation (2). In the Compton camera, SBP-based Compton images were reconstructed by superimposing Compton cones over all events. The projection of a cone onto a plane is a quadratic curve (an ellipse or hyperbola). In realistic imaging, the imaging space is discretized and the pixel centers rarely intersect the cones. Therefore, a technique was used to search for pixels that intersect the cone, and the pixels score the number of intersecting cones for reconstruction [191]. Because searching for pixels that intersect with the cone is computationally expensive, Wilderman et al. [192] reduced the computational complexity of the intersection calculation so that the SBP could be performed faster.

The angle estimation of the Compton scattering contained errors. In the case of ETCC, not only the scattering angle, but also the scattering plane can be estimated, and this error also needs to be considered. Therefore, back-projection is now performed as a probability distribution of the existence of the source, taking these errors into account, instead of scoring only the intersection of the cone and imaging space [191]. This method is also used as a back-projection in the FBP and EM methods.

If the object being measured is sufficiently far away from the Compton camera and the detector is sufficiently small, Compton imaging is similar to that of a pinhole camera. If the location of the scatterer is considered to be a point and back projection is made onto its celestial plane, the issues of projection angle and distance do not arise. However, in medical applications, contamination measurements in a plant, etc., where the object is relatively close and the 2D or 3D distribution needs to be reconstructed, Haefner et al. [193] successfully related 2D Compton 4π imaging to a 3D Radon transform with a known solution.

In the near-field case, the change in detection efficiency with the distance must also be considered. Takeda et al. [154] compared two methods with different probability density distributions: uniformly enlarged projection (UEP) and sampling equivalent projection (SEP). The SEP produced “ghost structures” around the image center. This is because many cones pile up near the field in front of the camera, which unfortunately leads to a misunderstanding of the source distribution unless the sensitivity is corrected. Conversely, the UEP avoided contamination owing to its differences in the probability density profile. Thus, they concluded that UEP is appropriate for near-field imaging. Example images of SBP are included in Figure 11 (left) and Figure 12b,c.

### 4.2. Filtered Back-Projection Methods

The image of the back-projection method is contaminated with noise and artifacts, although it is a fast and straightforward imaging method. In addition, the reconstructed images tend to be blurred because the source position and distribution are estimated by superposition of the probability distributions. Therefore, it is important to improve back-projection imaging methods, such as implementing a suitable filter, to reduce blurring. Daniel et al. [194] calculated the distance between the center of each pixel and the intersection in the search for the SBP. This method can be developed into FBP by setting the pixel values and thresholds corresponding to the distances. However, in another study by Lee et al. [195], an FBP imaging algorithm that overcomes the limitations owing to the azimuthal non-uniformity of the sources was reported. The developed algorithm significantly reduced the artifacts in the images compared to the conventional FBP method. Another important study on FBP using a spherical harmonic function was conducted by Basko et al. [196,197,198]. This method was improved by Parra et al. and Tomitani et al. [199,200,201] to account for the distribution of the scattering angles. In addition, Xu et al. [202] used PSF derived from a Monte Carlo simulation using the FBP method and demonstrated the excellence of the derived FBP imaging method. Recently, they implemented a Wiener filter in back-projection and investigated the power spectral density of the signal and noise to develop an appropriate SNR model for the restoration process [203]. The FWHM of ^228^Th source imaging with the proposed FBP was 9.8°, an improvement from 29° when using the simple back-projection imaging method. Thus, studies are being conducted to improve SBP with the application of FBP so that it could be useful for real-time imaging.

### 4.3. Expectation-Maximization Methods

The iterative image reconstruction approach consists of several choices, such as the basic representation of the object, basic physical modal of the detector, statistical model for measurements, and appropriate estimation criteria. The classical approach is the maximum likelihood (ML) estimation [204]. Wilderman et al. [205] in 1998 introduced the ML estimation approach to reconstruct the Compton images from event-based (list-mode) data. By passing through a series of improvements in iterative methods, both the bin-mode and list-mode for imaging near-field as well as far-field sources were studied. In 2014, Ikeda et al. [206] proposed two imaging algorithms based on a bin-mode estimation method: an accelerated EM method for the maximum likelihood estimation (MLE) and a modified expectation-maximization for maximum a posteriori (MAP) estimation. It has been reported that accelerated EM converges faster than the original EM algorithm, and the MAP estimate converges quickly into a sparse image. List-mode maximum likelihood expectation-maximization (LM-MLEM) has become a widely used method for Compton imaging [205]. A general equation for the MLEM method based on the list mode is illustrated in Equation (3).
(3)$$\lambda_{j}^{n}=\frac{\lambda_{j}^{n-1}}{S_{j}}\overset{N}{\underset{i=1}{\sum }}\frac{t_{ij}}{\sum_{k}t_{ik}\lambda_{k}^{n-1}}$$
where $\lambda_{j}^{n}$ represents the calculated amplitude of the image pixel j at the th iteration. N is the number of recorded events, and $t_{ij}$ is the weighted likelihood when event i originated from image pixel j. The sensitivity $S_{j}$ denotes the probability that a γ-ray originating from image pixel j is detected anywhere in the imager.

Another version of an iterative method is the ordered-subset expectation maximization (OSEM) method. OSEM works in the same manner as MLEM, but it uses data divided into subsets (portions of measured data), and updates are performed for each subset. If subsets are created appropriately, image convergence can be greatly accelerated with little degradation in the image quality [81,207,208].

An attempt to implement three-dimensional Compton imaging using MLEM was reported by Tronga et al. [209] in 2009. In MLEM, determining the parameter that corresponds to the response of the imaging system becomes the key point for reconstruction accuracy. However, the convergence speed of the adopted iterative reconstruction method was not sufficiently high because the imaging algorithms considered the high-energy γ-ray source located at a far distance, and the simulation programs developed at that time considered the simple detector system. Some studies have recently been conducted to improve the MLEM and compare it with other imaging models, with a focus on the application of Compton cameras in nuclear medicine and particle therapy [81,210,211,212,213]. Yabu et al. [214] reported the superiority of the LM-MLEM method over SBP using a Si/CdTe Compton camera. Tomographic imaging of ^111^In and ^131^I Ris within a tetrahedron-shaped phantom using SBP and LM-MLEM methods was demonstrated.

Recently, Yao et al. [215] reported a simulation and experimental study of 3D image reconstruction using the improved OSEM that considers the shift variant point spread function (LM-OSEM-SV-PSFs). It is claimed that the proposed method enhances the spatial resolution and reconstruction speed and weakens the effect of the spatial orientation of the imaging space, which causes distortion in the direction perpendicular to the detector plane.

Median root prior expectation-maximization (MRP-EM) is another iterative method. MRP is based on the one-step late algorithm, which modifies the OSEM by incorporating prior information [216]. This method assumes that the most probable value of a pixel is close to the local median of the image pixels in a neighborhood. The general equation of MRP-EM, by which the pixel value is calculated, is represented by Equation (4).
(4)$$\lambda_{j}^{\left(k,\;l+1\right)}=\frac{\lambda_{j}^{\left(k,\;l\right)}}{S_{j}\left(1+\beta \frac{\lambda_{j}^{\left(k,\;l\right)}-med\left(\lambda .j\right)}{med\left(\lambda .j\right)}\right)}\overset{N}{\underset{i\in S_{l}}{\sum }}\frac{t_{ij}}{\sum_{m}t_{im}\lambda_{m}^{\left(k,\;l\right)}}$$

Here in Equation (4), β is a hyperparameter that influences the degree of smoothness of the estimated images and med(λ.j) is the median of the image pixels over a neighborhood around the j th pixel. Sakai et al. [81] implemented this method in their studies to compare the imaging ability with other imaging modalities and reported that MRP-EM has the ability to produce high-quality images in a reasonable time. A median filter can reduce noise (especially spike noise) while preserving the contours. However, there is a problem that the sum of pixel values is not preserved. This may have affected quantitative evaluation. Many studies are being conducted to improve iterative methods, despite their intensive computation and ambiguous point conversion.

### 4.4. Stochastic Origin Ensemble Methods

The stochastic origin ensemble (SOE) method was introduced by Andreyev et al. [217,218,219]. The imaging algorithm is based on the Monte Carlo Markov chain algorithm, which assumes that each event is associated with a pixel of the reconstructed image. With respect to the transition probability, each event may move to another pixel for each iteration. For the initialization of this iterative method, the possible origin points and their new location within a back-projected cone are randomly selected. Finally, the pixel values of the former and new candidate points are compared in accordance with the acceptance probability; the former point or the current point will be the new point. The image was reconstructed using the estimated new location for all the events. Figure 14 shows an example of an implication of the event position by the SOE method, where the event may move to the new location or remain at the current (old) location, accordance to the acceptance probability equation [207]. The general equation for the position transition is represented by Equation (5).
(5)$$p_{i}=\min\left(1,\;\frac{{\left(d\left(n_{i}\right)+1\right)}^{d\left(n_{i}\right)+1}{\left(d\left(o_{i}\right)-1\right)}^{d\left(o_{i}\right)-1}}{d{\left(n_{i}\right)}^{d\left(n_{i}\right)}d{\left(o_{i}\right)}^{d\left(o_{i}\right)}}\right)$$
where for the *i*th former event, $n_{i}$ is being the new selected position, $o_{i}$ is being the old position, and the transition probability is given by $p_{i}$. Sakai et al. [81] used this imaging algorithm as an iterative algorithm without ML-EM and compared it with other methods. It was reported that the SOE reconstructed images more quickly for an even higher number of iterations.

## 5. Future Perspectives

Since the first developed Compton camera, it has undergone many development phases in terms of detectors, signal-processing hardware, imaging algorithms, and applications. At this stage, research on the development of Compton cameras for medical applications is progressing at a good pace. Han et al. made some statistical comparison of the Compton camera with the collimated detectors [50]. The results showed that the Compton camera outperforms the collimated anger camera when evaluated for an equal number of detected events. With the Compton camera, the spatial resolution for a 26 cm diameter disk object was imaged with a resolution better than 12 mm FWHM. Moreover, the sensitivity was 15–20 times higher than the collimated camera. Similarly, Fontana et al. compared the radial event distribution, detection efficiency, and the imaging quality of a commercial SPECT-anger camera with the Compton camera by means of Monte Carlo simulations [51]. The result showed that the detection efficiency of the Compton camera was increased by the factor larger than an order of magnitude with an enhanced spatial resolution for energies beyond 500 keV. From such studies, it is believed that the Compton camera could be a new option in nuclear medicine in the near future. However, we are not able to achieve the desirable image resolution that is useful for medical imaging. The spatial resolution is insufficient, especially in the medical field, and further improvements are required in terms of detector performance, data acquisition and processing hardware, and imaging algorithms.

The detector performance required for Compton cameras is a high energy resolution, high measurement position accuracy, large reaction cross-section, small Doppler effect, and fast response time. Semiconductor detectors generally have a high energy resolution and measurement position accuracy. In terms of the reaction cross-section, the appropriate material depends on the energy to be measured. When measuring low-energy γ-rays (less than a few hundreds of keV), the scatterer should be made of a material with a small absorption cross-section. The Doppler effect is also a significant factor and can be reduced using a Si or a C [57] detector. Conversely, when measuring MeV γ-rays, the Doppler effect is negligible, and the reaction cross-section is too small for detectors with low atomic numbers.

In a Compton camera, it is necessary to know at least two measurement sets of the position and energy to measure a single event. Many high-precision analog to digital (AD) conversions are required. The radioactivity of the measurement target is particularly high for medical applications and plant environmental measurements. Thus, it is necessary to have a system to prevent the equipment from pile-up, owing to the high frequency of the γ-rays. The ratio of dead time to live time in the Compton camera depends on the electronic circuit implemented in the Compton camera. Therefore, improvements in the ASIC would enhance the performance of Compton cameras [220]. If the desired γ-ray energy is determined, it will be possible to devise a layout that reduces the measurement of unnecessary scattering. Moreover, as reported in the Iyomoto et al. [221] the implementation of a suitable technique such as a position-sensitive transition-edge sensor (TES) to minimize the negative influence of Compton scattering could lead to improvements in Compton cameras. 

Another factor intended for further development in Compton cameras is the image reconstruction speed as well as the data acquisition speed. As explained in Section 4, a rapid image reconstruction method is required. Compton imaging in tomography reconstructs images of a large number of events. Moreover, data processing tends to be both complex and computationally intensive. The EM method is the mainstay of Compton camera research; however, it requires additional computational resources. Algorithms must be optimized to perform imaging at realistic times and machine specifications for each objective. Currently, the LM-MLEM imaging algorithm is widely used and is under further improvement. Although the current stage of the iterative method is acceptable in nuclear medical imaging, it is not yet realizable for the real-time imaging of γ-rays. An alternative to LM-MLEM is the FBP algorithm; however, the image resolution is insufficient and needs to be improved significantly. To improve the imaging algorithm, it is necessary to determine the characteristics of the object to be measured.

Nevertheless, a detailed analysis of PSF can improve imaging algorithms [222]. The PSF of Compton imaging varies depending on the configuration of the Compton camera and target energy. Therefore, it is important to conduct specific research for each application and Compton camera configuration, along with general Compton camera research.

## Figures and Tables

**Figure 1 sensors-22-07374-f001:**
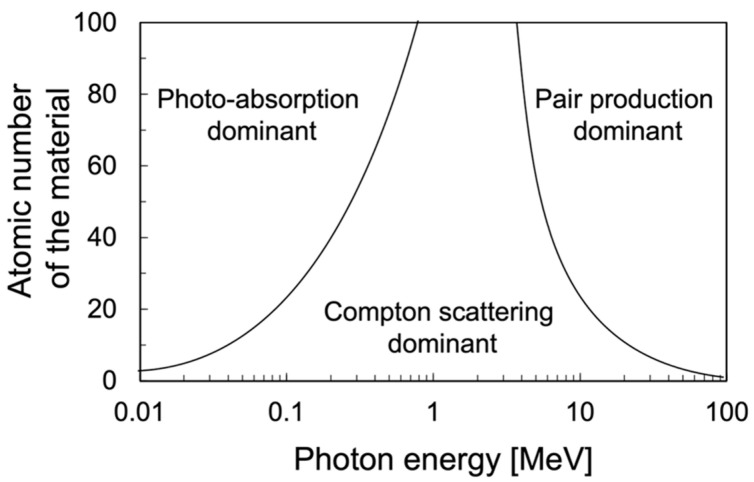
Energy at which photoelectric effect, Compton scattering, and pair production are dominant with respect to atomic number (Z) of the absorber.

**Figure 2 sensors-22-07374-f002:**
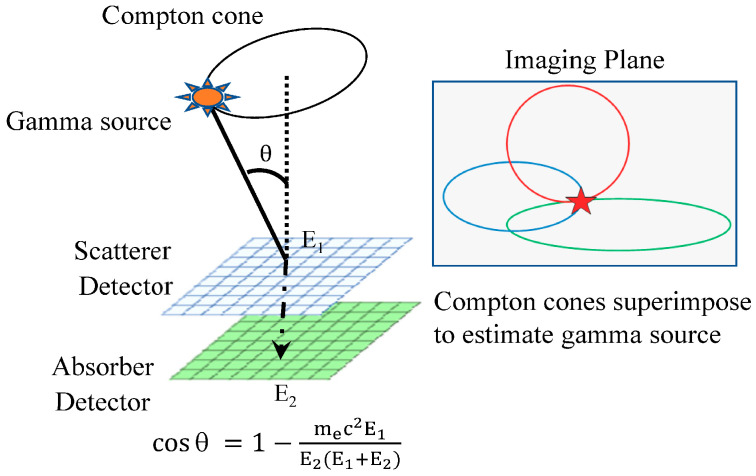
Schematic illustration of a general Compton camera (**left**); Compton cones of each event are superimposed to locate the γ-ray source (**right**).

**Figure 3 sensors-22-07374-f003:**
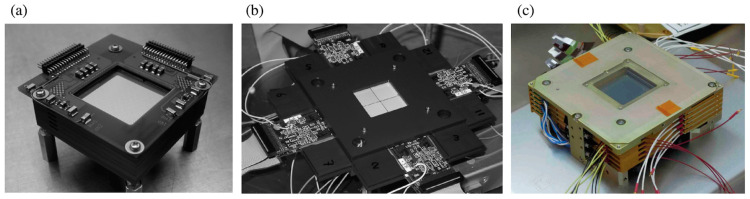
Photographs of (**a**) multilayered DSSDs stack [65], (**b**) the CdTe detectors stack [65], and (**c**) the prototype Si/CdTe Compton camera [66]. Reproduced with permissions from Watanabe et al. [65] and Takeda et al. [66].

**Figure 4 sensors-22-07374-f004:**
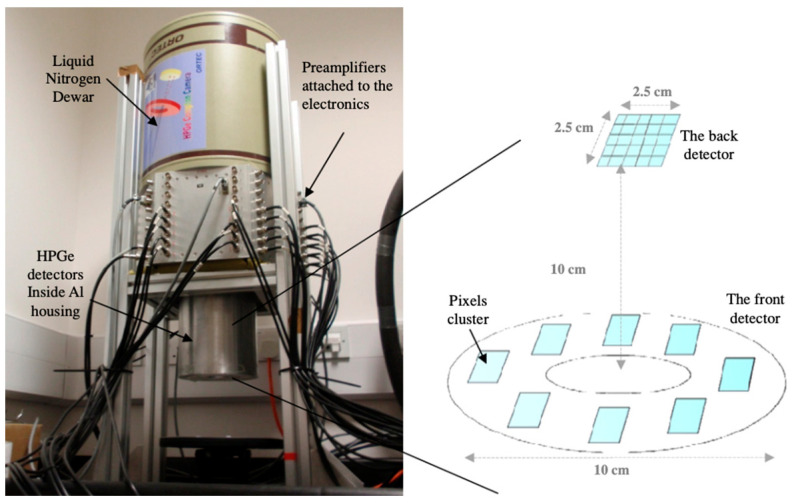
Pixelated HPGe Compton camera with the detector configurations [109]. Reprinted with permissions from Alnaaimi et al. [109].

**Figure 5 sensors-22-07374-f005:**
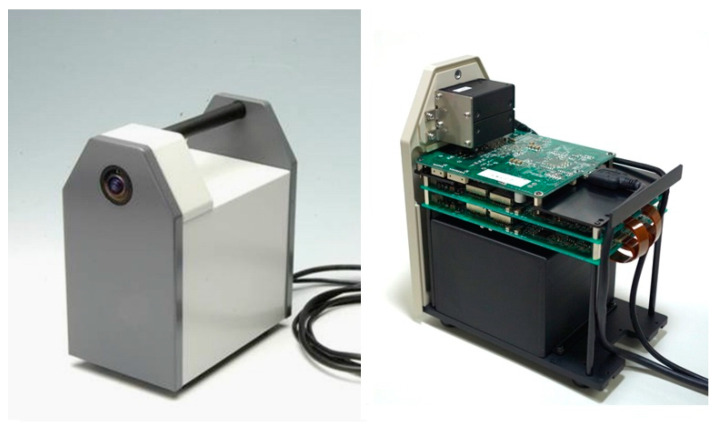
Ce:GAGG and MPPC based DOI handy Compton camera developed by Kataoka et al. [118]. Left photograph shows the Compton camera in compact form and the right picture shows the internal structure of the camera. Reprinted with permissions from Kataoka et al. [118].

**Figure 6 sensors-22-07374-f006:**
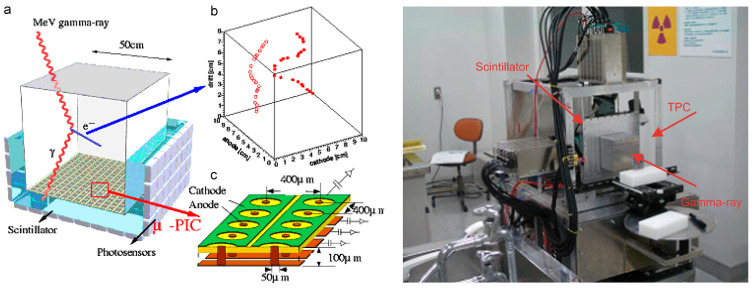
Conceptual structure (**left** (**a**) ETCC (**b**) tracks of electrons and (**c**) schematic structure of micro-pixel gas chamber) and actual photograph of ETCC (**right**) developed by Kabuki et al. [129]. Reproduced with permissions from Kabuki et al. [129].

**Figure 7 sensors-22-07374-f007:**
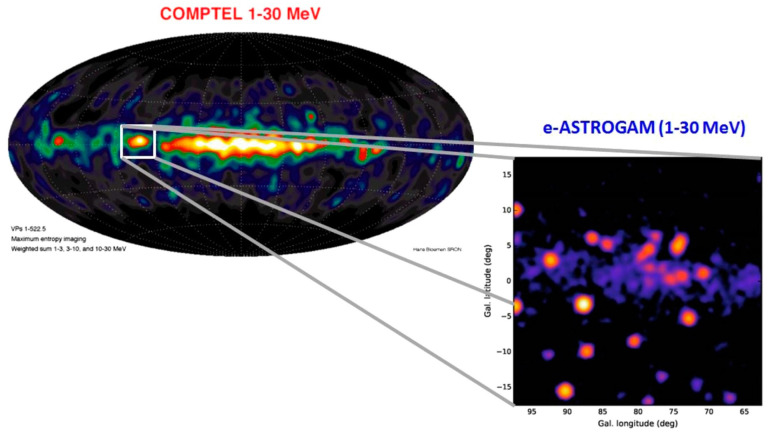
1–30 MeV gamma in the sky as observed by COMPTEL in the 1990s (**upper left**) and the simulated Cygnus region in the 1–30 MeV energy region expected from e-ASTROGAM (**lower right**) [151]. Reproduced with permissions from Angelis et al. [151].

**Figure 8 sensors-22-07374-f008:**
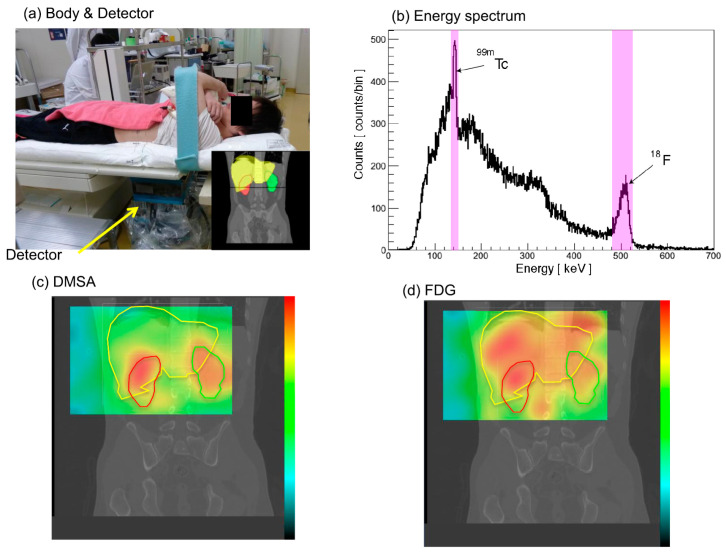
(**a**) Human experiment setup using Compton camera. (**b**) Energy spectrum obtained in the study. (**c**) Compton image overlaid with CT image for ^99m^Tc DMSA radiopharmaceuticals. (**d**) Compton image overlaid with CT image for ^18^F FDG radiopharmaceuticals. All the images are reprinted from Nakano et al. [84]. Reprinted with permissions from Nakano et al. [84].

**Figure 9 sensors-22-07374-f009:**
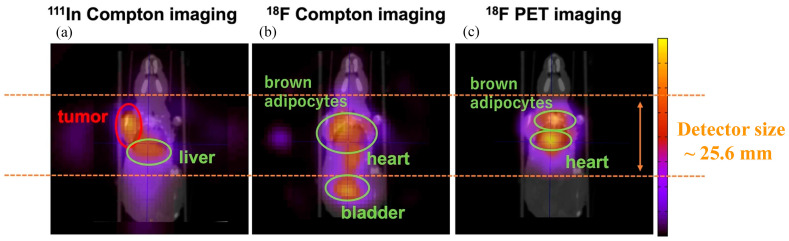
In vivo Compton imaging of accumulation of (**a**) ^111^I in mouse tumor and liver, (**b**) ^18^F-FDG in heart and bladder, and (**c**) The PET imaging of ^18^F-FDG accumulated in mouse heart. The images were superimposed with the CT images [156]. Reproduced with permissions from Uenomachi et al. [156].

**Figure 10 sensors-22-07374-f010:**
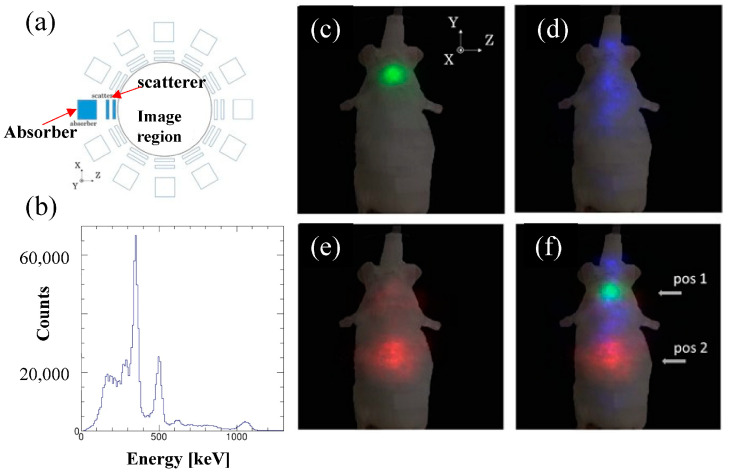
(**a**) Configuration of multi-angle data acquisition measurement of Ce:GAGG Compton camera. (**b**) Energy spectrum obtained via 10-min measurements from an angle in a mouse injected with ^131^I, ^85^Sr, and ^65^Zn. (**c**) Compton image of ^131^I, (**d**) ^85^Sr, and (**e**) ^65^Zn, and (**f**) fused images of all three tracers. All the figures are reprinted from Kishimoto et al. [155]. Reproduced with permissions from Kishimoto et al. [155].

**Figure 11 sensors-22-07374-f011:**
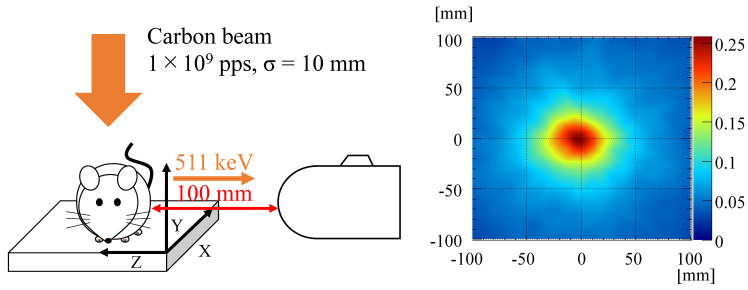
In vivo real-time monitoring of annihilation γ-rays generated by CIRT using Si/CdTe Compton camera. Experimental setup (**left**). Compton image of 511 keV annihilation gammas (**right**). Images have been rearranged and reprinted from Shiba et al. [87].

**Figure 12 sensors-22-07374-f012:**
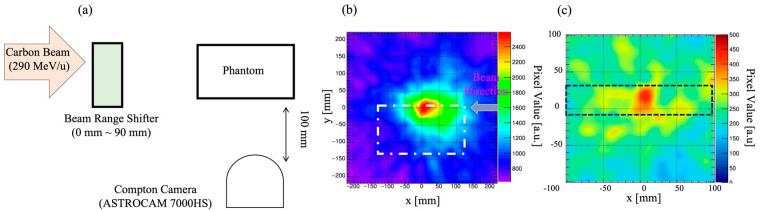
On-beam range monitoring using Si/CdTe Compton camera in CIRT. (**a**) Experimental setup. (**b**) Compton image of 511 keV annihilation gammas [85]. (**c**) Compton image of 718 keV prompt gammas [86]. Dotted line in figures (**b**,**c**) represents the phantom periphery. All the figures are rearranged and reprinted from Parajuli et al. [85,86].

**Figure 13 sensors-22-07374-f013:**
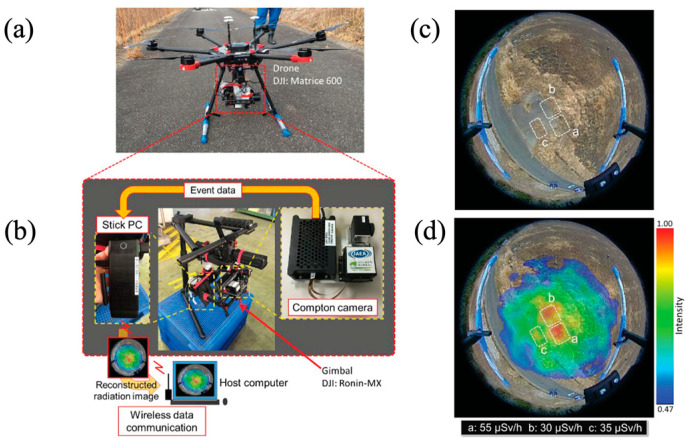
(**a**) Drone used for carrying Compton camera for radiation monitoring. (**b**) Zoomed view of radiation measurement system. (**c**) Optical image taken using drone at altitude of 6 m. (**d**) Radiation image measured using the drone system in which the a, b, and c regions represent the hotspots [181]. Reproduced with permissions from Sato et al. [181].

**Figure 14 sensors-22-07374-f014:**
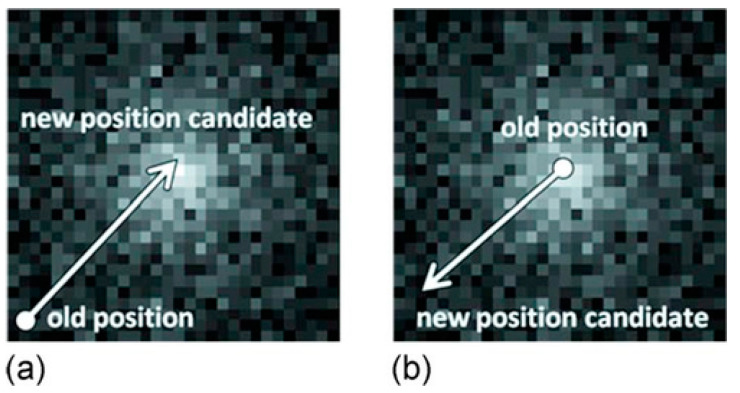
Example of an implication of event position by the SOE method. The event may move to the new location or remain at the current (old) location, according to the acceptance probability equation: (**a**) a new position will be accepted with 100% probability as in this case the event density increases; (**b**) the new position will be accepted with low probability due to the sharp decrease in event density. The figures are rearranged and reprinted with permissions from Andreyev et al. [218].

**Table 1 sensors-22-07374-t001:** Some RIs released into the atmosphere during the Fukushima nuclear accident in 2011.

Leakage Isotopes from FDNP	Emission Amount [PBq]	Disintegration	Decay Time
^137^Cs	6.1~35.9	β^−^	30 years
^134^Cs	11.8~18	β^−^	2.1 years
^85^Kr	44	β^−^	10.76 years
^129^I	5.5 × 10^−5^~5.5 × 10^−6^	β^−^	1.57 × 10^7^ years
^131^I	65~380	β^−^	8 days
^133^Xe	11,400~15,000	α β^−^	5.245 days

**Table 2 sensors-22-07374-t002:** Properties of some semiconductor detectors widely used in Compton cameras.

Semiconductor Detector	Density [g/cm^3^]	Atomic Number [Z]	Band Gap Energy [eV]	Ionization Potential (∊) [eV]
Si	2.33	14	1.12	3.6
CdTe	5.58	48, 52	1.44	4.43
Ge	5.33	32	0.67	2.9
CdZnTe	5.81	48, 30, 52	1.6	1.6
HgI_2_	6.40	80, 53	2.13	4.2
GaAs	5.32	31, 33	1.42	4.3

**Table 3 sensors-22-07374-t003:** Properties of some scintillator detectors used in Compton cameras.

Scintillators Detectors	Density [g/cm^3^]	Light Yield [photon/MeV]	Decay Time [ns]	Peak Emissions [nm]	Atomic Number (Z) [eV]
NaI(Tl)	3.7	45,000	230	415	51
CsI(Tl)	4.5	56,000	1000	530	54
Ce:GAGG	6.6	57,000	88 (91%) + 258 (9%)	520	54.4
CaF_2_(Eu)	3.18	24,000	940	435	54
BGO	7.13	8000	300	480	74
Ce:LaBr_3_	5.1	75,000	30	375	46.9
Ce:LSO	7.35	25,000	42	435	66

**Table 4 sensors-22-07374-t004:** Commonly used RI tracers in nuclear medicine with their properties.

Radio Isotopes	Half-Life Time	Decay Type	Energy [keV]
^111^In	2.83 days	EC	171, 245
^123^I	13.2 hrs	EC	159
^99^Tc	6.0 hrs	EC	141
^18^F	108 min	An. (β^+^)	511
^67^Ga	78.3 hrs	EC	93, 184, 296, 388
^85^Sr	64.8 days	EC	514
^64^Cu	12.7 hrs	EC	579, 653, 1350
^131^I	8.04 days	EC	364
^65^Zn	244 days	EC	1116

EC: Electron capture, An.: annihilation.

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
