# Peer review of "Development and Applications of Compton Camera—A Review"

_sensors, 2022, doi:10.3390/s22197374_

Round 1
Reviewer 1 Report
Thank you for reviewing Compton camera technology with many references.
Reviewer 2 Report
The review is very interesting and well organized. I have the following minor comments:
1) Figure 1 is taken from the book by Glenn Knoll (Radiation Detection and Measurements, Wiley). Please carefully check the copyright.
2) Most of the citations are to papers by Japanese groups. I recommend to extend bibliography including European contributions on gamma ray spectrometer based on SiPM:
· page 4: when discussing SPECT you can mention also multimodal imaging (i.e. combination of SPECT with MRI) and cite https://doi.org/10.1109/TRPMS.2019.2951355 as an example of SPECT based on SiPMs
· section 3.4: when discussing environmental monitoring, you can cite these gamma cameras based on SiPM coupled to scintillators: 2” NaI ( https://doi.org/10.3390/s22041412 ) also flying on a drone and 3” LaBr3 ( https://doi.org/10.1109/JETCAS.2020.3029570 ) including embedded machine learning for direction sensitivity.
3) Page 5, define DSSD
4) Section 4: discuss the computational cost of the discussed methods and clarify if image reconstruction is performed off line or in real time (and on which hardware platform).
5) Section 5: prove some quantitative summary comparison of Compton cameras vs collimator detectors. Discuss how the spatial resolution, especially in the medical field, could be improved.
Reviewer 3 Report
This review paper tries to summarize the history and recent progress of Compton imaging.
Generally, the paper is well written, however, citation is a bit biased to the specific working groups and some references are lacked. You should cite more papers with wider aspect.
Here are detailed comments.
ll73 “that can efficiently these polarized γ-rays” a verb is lacked.
Table1 PBq
ll325 Regarding solid detector based electron tracking Compton camera, there are more works, such as
Wen, Jiaxing, et al. "Optimization of Timepix3-based conventional Compton camera using electron track algorithm." Nuclear Instruments and Methods in Physics Research Section A: Accelerators, Spectrometers, Detectors and Associated Equipment 1021 (2022): 165954.
Yoshihara, Y., et al. "Development of electron-tracking Compton imaging system with 30-μm SOI pixel sensor." Journal of Instrumentation 12.01 (2017): C01045.
ll 503 I think you are citing the wrong paper, it should be [149], NOT [148] for Compton-PET
Figure 9 seems to be also wrong. [149] shows the simultaneous visualization of 111In and 18F with PET and Compton imaging. Please replace the figure.
[148] is not Compton-PET, just a Compton camera.
ll679 for Fukushima related applications, there are earlier works to be referenced, such as,
Shikaze, Yoshiaki, et al. "Field test around Fukushima Daiichi nuclear power plant site using improved Ce: Gd3 (Al, Ga) 5O12 scintillator Compton camera mounted on an unmanned helicopter." Journal of Nuclear Science and Technology 53.12 (2016): 1907-1918.
Jiang, Jianyong, et al. "A prototype of aerial radiation monitoring system using an unmanned helicopter mounting a GAGG scintillator Compton camera." Journal of Nuclear Science and Technology 53.7 (2016): 1067-1075.
There will be other Compton imaging systems to be mentioned.
multi-pixel CdTe based Compton camera
Tomita, Hideki, et al. "Gamma-ray source identification by a vehicle-mounted 4π Compton imager." 2020 IEEE/SICE International Symposium on System Integration (SII) . IEEE, 2020.
Medipix single layer CdTe based CC
Turecek, D., et al. "Single layer Compton camera based on Timepix3 technology." Journal of Instrumentation 15.01 (2020): C01014.
GAGG Omnidirectional
Takahashi, Tone, et al. "Development of omnidirectional gamma-imager with stacked scintillators." 2013 3rd International Conference on Advancements in Nuclear Instrumentation, Measurement Methods and their Applications (ANIMMA) . IEEE, 2013.
some groups mention to use TES for Compton imager
Iyomoto, Naoko, et al. "Development of gamma-ray position-sensitive transition-edge sensor microcalorimeters." Journal of Low Temperature Physics 200.5 (2020): 233-238.
Round 2
Reviewer 3 Report
All the previous comments were well addressed and the manuscript seems to be ready for publication.